# Chatbot Service Quality: An Experiment Comparing Two Countries with Different Levels of Digital Literacy

**Julio Vena-Oya** [1,*] **, José Alberto Castañeda-García** [2] **and Jan Burys** [2]

1    Department of Business Administration, Marketing and Management, University of Jaén, 23071 Jaén, Spain
2    Department of Marketing, University of Granada, 18010 Granada, Spain; jalberto@ugr.es (J.A.C.-G.)
*    Correspondence: jvena@ujaen.es

**Abstract:** The use of new technologies in tourism is bringing about a genuine revolution in the sector, where automated services, such as chatbots, are increasingly being used to perform some of the tasks involved in service delivery. However, the emergence of new technologies in a sector as globalized as tourism can mean that not all users are able to perceive the benefits of these innovations. Therefore, the aim of this study is to determine whether the digital literacy (DL) of a tourist may lead to different evaluations of the quality of the electronic service they received, both overall and as per the different dimensions of quality. This study compares a more innovative and interactive service-provision technology (a chatbot) with a more traditional one (email). To this end, an experiment was conducted in which 124 participants from Spain (higher DL) and the Czech Republic (lower DL) were asked to interact with these two technologies in a simulated hotel customer-service scenario. The results show that individuals with higher DL rated the quality of service received via chatbot higher than individuals with lower DL. The latter prefer email because they perceived it to provide greater security, empathy, reliability, and information quality. However, participants with higher DL preferred the chatbot due to its greater competence in completing the task. Finally, the participants rated the responsiveness of the chatbot higher than that of email. These results can help the introduction of chatbot-based customer service in the tourism sector.

**Keywords:** technology acceptance; digital literacy; e-service quality; automated services; technology in tourism

## 1. Introduction

Today's so-called "fourth industrial revolution" is characterized by the rise of technologies related to robotics and artificial intelligence (hereinafter, AI), among others [1]. This shift is not only changing our consumption habits and lifestyles but is also affecting how we interact [2,3]. Tourism is not immune to this reality [4]; indeed, as part of the fall-out from the global COVID-19 pandemic, this transformation has accelerated in the sector [5], with certain tourism firms already beginning to adopt such technologies [6].

The adoption of new technologies in tourism service provision is leading to cost reductions in a number of areas, as well as an increase in the efficiency of certain services [7,8]. Furthermore, the tourism sector also benefits from the growing use of technologies that support customer service [5,9], which has led to a transformation in delivery [10], affecting the consumer experience [5]. However, for both users and firms, adopting a new technology is far from simple, and its introduction must be carefully planned if it is to be accepted by consumers [11–13].

From the consumer perspective, the adoption and evaluation of such technologies vary, depending on the individual [14]. Users' level of digital literacy, in particular (hereinafter, DL), is one of the key factors [15,16], and it plays an even more critical role in the case of advanced technologies, such as chatbots. Thus, the individual's level of DL will affect their behavioral patterns in terms of how they make use of information, their

technology acceptance [13,17], their use intention [18], the perceived usefulness of the technology [12,19], and its fulfillment of user expectations [20,21]. It is this latter aspect on which the present study focuses its attention: on determining whether users (tourists) are equipped to adopt this type of technology, taking into account the effect of their DL on their evaluation of a chatbot's perceived service quality.

To this end, an experiment was conducted into tourists' evaluation of two technologies that are used in the provision of online hotel services, comparing a traditional tool (email) and a more recent and innovative one (chatbot). This evaluation was performed on a set of six dimensions of online service quality by a sample of tourists from two different countries, one of which is characterized by a level of DL below the EU average (Czech Republic) and the other above that average (Spain). This approach is intended to help firms to work toward the effective implementation of this type of technology in the sector by taking into account the varying characteristics of their [12].

This type of study has been carried out in other sectors prior to introducing a given technology, as in the case of online banking [11,22] or other robots in tourism [23]. However, this type of technology is highly novel in the tourism sector and there are not yet many firms implementing it [6]. If we take into account that this technology is increasingly present in the tourism sector [3] and that more and more companies are opting to use it in different services offered to their customers, such as making or modifying reservations, which means a reduction in costs for the company and time for the tourist [9], it is necessary to know the users' opinion on the matter. For this purpose, some authors have been using some technology acceptance models, such as the UTAUT model, to find out whether they are willing to use it [24]. These works on technology acceptance use digital literacy to analyze the level of technology acceptance; however, a different level of digital literacy may also imply different perceptions of perceived quality, which may further hinder acceptance in certain regions [25,26]. However, this is the first work, to our knowledge, that attempts to determine the factors affecting the introduction of chatbots in the tourism sector by studying the quality of service offered by an established technology such as email versus a new one such as a chatbot. This can mean cost savings for companies but also time savings for tourists, thus increasing their satisfaction with the destination [27]. Thus, firstly, the scientific novelty of this work lies in the use of the perception of quality of service in countries with different levels of digital literacy (Spain vs. Czech Republic). This represents an advance in terms of understanding how these systems should be correctly implemented in a sector as important for the economy of countries as tourism, given that, as not all countries perceive it as better (the Czechs see some aspects of email as better), this implementation should not be generalized and managers should be staggered according to the digital readiness of citizens and tourists in each country.

## 2. Literature Review

### 2.1. The Use of Chatbots in the Tourism Sector

In certain contexts, service automation can deliver superior performance compared to the human touch, generating greater service user satisfaction. For example, Ref. [28] found that chatbots do not lose interest in the conversation if it lasts longer than usual, unlike human operators. Some consumers even show interest in communicating with a bot simply because it offers a novel experience. In addition, these systems can generate a sensation among users of being able to keep in touch with the firm around the clock [29], which is accentuated even further if the information being offered is more relevant and accurate than human personnel could provide [30].

However, despite the many benefits of automating services [8], the loss of human contact can also potentially generate a sense of rejection toward the technology, especially if the individual is not sufficiently skilled to use it properly [31]. In addition, privacy concerns continue to be an important factor that exerts a chilling effect on the adoption of different technologies [32].That said, despite only being in an early phase of development, chatbots

have started to be used by firms such as Makemytrip, Expedia, Kayak, Skyscanner, and Cheapflights, among others, for customer support.

Yet, despite the pertinence of exploring the use of chatbots in tourism, studies dealing with this specific topic are still scarce [33]. The few that do take this perspective primarily tend to address tourist use intention [34] and the capabilities and possibilities of this technology [35], and there are very few causal studies that analyze the impact of automated services on the tourist experience [36].

Among the few findings to date, it has been identified that if chatbots fulfill certain quality requirements, customers derive a satisfactory experience from interacting with them [37]. Hence, it is necessary to better understand the tourist's assessment of service quality when engaging with chatbots [38].

### 2.2. The Dimensions of Quality in Service Delivery

Ever since [39] established the different dimensions of service quality, their SERVQUAL scale has been used extensively, including for online services (e.g., [22,40]).

In terms of virtual environments, authors such as [41] seek to unify the "traditional" dimensions of service quality [42] with others specific to the online context, focusing on the perspective of the service provider. Such works identify that there are up to eight dimensions: website design, reliability, security, responsiveness, fulfillment, personalization, information, and empathy. Other authors, such as [22,43], who also follow the model proposed by [39] determine that the dimensions that make up electronic service quality (ESQ) are security, empathy, reliability, and responsiveness. Other dimensions identified in the literature are information quality [44] and competence [44,45].

Of these, security is one of the most important dimensions, since the consumer may be reluctant to provide personal information when using a digital service [32]. It is clear, then, that this factor is decisive when it comes to generating greater satisfaction and thereby increasing customer acceptance of chatbot technology, especially given that it is still at an early stage in its development [11,13].

Another key factor that determines the quality of a service is empathy. This is particularly so in the case of digital services, in which the user interacts with a machine [31], where empathy has been identified as a fundamental ingredient in the consumer's satisfaction with a given technology [22].

Reliability is used in numerous studies referring to the correct performance of tasks that have been assigned to the system [44,46]. This dimension has also been used to analyze consumer satisfaction and the degree of technology acceptance [11,22]. In a similar vein, the competence dimension is also found to be significant [47] albeit more closely related to system availability [45] and efficiency [46,48].

A further factor that attracts relative consensus in the literature is responsiveness, which is linked to the immediacy of service responses. This dimension appears to be widely justified in the literature [47–49]. Closely linked to this dimension is information quality. In digital systems, not only is immediacy an important factor, but the quality of response must also be in line with what the user expects—that is, it must fulfill the expectations that the system may generate [30]. Information quality will drive-up use intention and, in turn, facilitate the correct implementation of the application technology in the sector [12].

In sum, most of the literature proposes the six dimensions outlined in Figure 1 as those that make up ESQ:

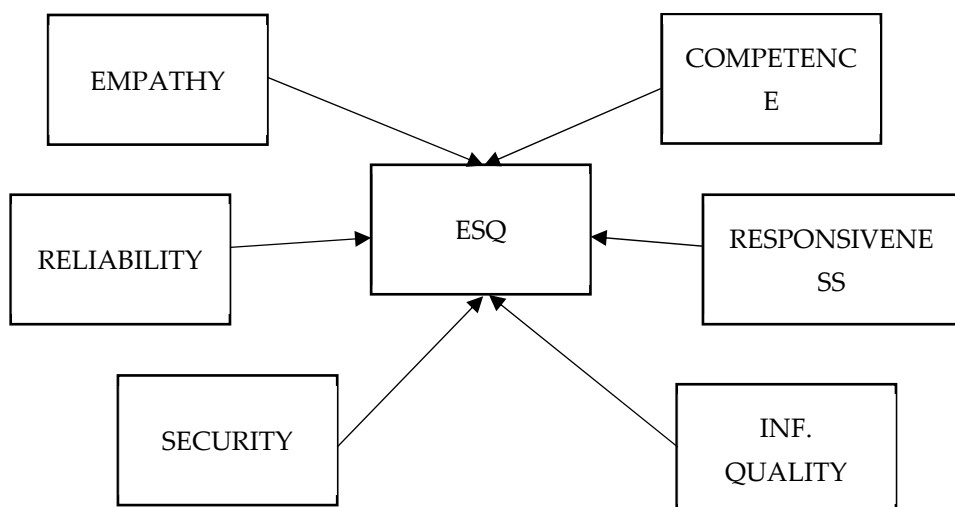

**Figure 1.** Dimensions of ESQ.

*2.3. Digital Literacy*

When it comes to the adoption of a new technology, DL plays a crucial role [19]. DL was defined by [50] as "the ability to understand and use information in multiple formats from a wide range of sources when it is presented via computers", this term being used also in relation to technology adoption [51]. DL has been shown to affect behavioral patterns in the use of information and technology [52], and some authors have demonstrated how a high level of DL engages individuals in the use of technology in different contexts (such as education, health, or business) [53]. It has also been found that simply having access to technology does not guarantee that it will be well used or that the expected result will be achieved if the individual does not possess the necessary skills to use it [19].

In the tourism sector, some studies have used this construct to explain the perceived ease of use of some technologies in their travels [54] by including technology acceptance models [55], linking the DL with the motivation to obtain autonomy during travel [56] or to explain the perception of the congruency between digital app presentations and their actual experiences [57]. DL has been used over the years within these technology acceptance models, such as TAM or UTAUT in the tourism sector [57].

Turning to the broad sector of information technology, it has been shown that DL is key to users' acceptance of particular technologies [13,18] and also increases use intention [12,18]. Furthermore, a high level of DL will increase perceived ease of use and perceived usefulness [13,19] but it will also exert an effect on expectations regarding the performance of the technology in question or on the hedonic motivation to use it [21].

It is to be expected, then, that DL may affect the perceived quality of a system, given that a low level of DL can affect not only use intention [16,31], as noted, but also the technology's perceived ease of use, due to the user's lack of skill [58] and, consequently, its perceived usefulness [12]. This lack can lead to frustration and thus dissatisfaction, rendering the user unable to properly appreciate all the possibilities that a new technology can offer [59].

Finally, several authors have determined that digital skills influence not only use intention but also the user's perception of automated services [60] This means that those with low DL may even find such services useless [61], while those with a high level of DL present a more positive attitude toward them [58,62]. DL, then, can ultimately affect users' trust in a technology [13].

In short, the level of DL will affect the behavioral patterns they present in relation to a new technology [13,52]. This will affect, in turn, not only their technology acceptance and use intention [18] but also the technology's perceived usefulness. And, in general, a higher level of DL will lead them to present a more positive attitude when dealing with a new technological challenge [19,58].

Therefore, the following hypothesis is proposed in relation to DL and ESQ:

H1: DL moderates the user's perception of the perceived quality of a new technology compared to a mature one.

H1A: the perception of security will differ from a new technology compared to a mature one.

H1B: the perception of empathy will differ from a new technology compared to a mature one.

H1C: the perception of reliability will differ from a new technology compared to a mature one.

H1D: the perception of competence will differ from a new technology compared to a mature one.

H1E: the perception of responsiveness will differ from a new technology compared to a mature one.

H1F: the perception of information quality will differ from a new technology compared to a mature one.

## 3. Methodology

### 3.1. Experiment Design

To fulfill the research aim—which required a comparison of two technologies used in the provision of online services (email vs. chatbot) and an analysis of the differences identified comparing two countries characterized by different levels of DL—a full ($2 \times 2$) factorial design was implemented. The sample comprised 124 university students who were recruited during the first half of 2021 (60 from the Czech Republic and 64 from Spain) from their respective universities. All participants consented to participate in this study. These two countries were selected on the basis that, given the European Union average of 26.5% of the population having "above basic" digital skills, the population of the Czech Republic presents a value below that average (24.1%), and Spain, above (38.1%).

To complete the experimental design, a hypothetical situation was created for a fictitious establishment—Hotel Gu—where the participant had a reservation and wanted to make a room change. The sample was divided into two groups (one that would make the change via email and the other via a chatbot), and each group comprised a mixture of higher-DL (Spanish) and lower-DL (Czech) participants. Each individual was given the corresponding guide, according to which technology they were using. The guide outlined the scenario and provided instructions for completing the task.

The chatbot was implemented and the decision tree created using the Chatfuel service (https://chatfuel.com, accessed on 20 March 2024), which was interfaced with the hotel's fictitious Facebook profile. This service also enables chatbots to be interconnected with websites, Instagram, and other social networks. The chatbot, which was purpose-designed for this experiment, was capable of showing images of the different room options and presented different buttons that the participant could click to confirm the final booking instantaneously, evaluate the service received, and personalize the conversation using their name for the duration of the interaction. This type of chatbot—created specifically to fulfill a narrow range of tasks—is one of the most popular alternatives, thanks to its low cost compared to more sophisticated chatbots that can sustain more diverse conversations with users.

Meanwhile, those participants who interacted by email also received options for modifying their hotel booking, this time in a personalized message containing images of the alternatives. These participants were required to wait up to an hour to receive a response to their email request.

Participants in both groups were able to complete the task with no difficulty. Upon successful completion of the task, the individual was redirected to a questionnaire to evaluate their experience. This asked about their experience of the customer service received through the medium in question, different socio-demographic questions, their

degree of experience in making online accommodation bookings, and their engagement with the task presented to them in this study.

### 3.2. The Sample

The total sample of 124 undergraduate and Master's students who participated in the experiment was divided into 60 for the low-DL group and 64 for the high-DL group. DL (https://tinyurl.com/4y6btwe6, accessed on 20 March 2024) was accounted for using Eurostat data, which show that 57% of young Spanish people (between 16 and 24 years of age) have above-average digital skills compared to only 42% of Czechs. The decision to choose this segment of the population was due to the low levels of DL presented by other (older) audiences, which could have hampered their ability to correctly perform the tasks assigned to them. As participants were required to complete the tasks without the assistance of the interviewers, this could have frustrated some individuals and thus skew the results of the study. The data profile of the sample is shown in Table 1: 60.5% were women, and the average age of the participants was 24. These data were balanced across the two technologies under analysis, with no differences in either the mean age (*p*-value = 0.54) or in the percentage of women (*p*-value = 0.58) (where H0 means that there are no differences in the average by groups). By country, no differences were identified in the socio-demographic variables, with *p*-values of 0.69 and 0.11, respectively, for age and gender.

**Table 1.** Sample data.

|  | **Czech Republic** | **Spain** | **Total** |
|---|---|---|---|
| Gender (female) | 32 (53.33%) | 43 (67.19%) | 75 (60.48%) |
| Gender (male) | 28 (46.67%) | 21 (32.81%) | 49 (39.52%) |
| Age (std. dev.) | 25.57 (8.38) | 22.03 (1.89) | 23.74 (6.22) |
| Technology (email) | 30 (50%) | 32 (50%) | 62 (100%) |
| Chatbot | 30 (50%) | 32 (50%) | 62 (100%) |

### 3.3. Measurement Scales

Based on the dimensions of ESQ identified in the literature, the questions for measuring each construct were formulated. From the work of [41] the items corresponding to reliability, information quality, empathy, and responsiveness were extracted and adapted. The items on the security scale were adapted from [63]. To operationalize the competence dimension, items from [41,47] were combined. Finally, to measure the overall evaluation of service quality, the items used by [64] were adapted to the present study.

In all cases, 5-point Likert scales were employed, in which 1 indicated "completely disagree" and 5 indicated "completely agree", and the Cronbach's alpha value was above 0.7 (see Supplementary File S1 for all items and the internal consistency indicator, mean, and standard deviation).

Finally, to avoid the bias that might have arisen in the responses due to the country of origin of the respondents, a rescaling technique was carried out to standardize the response. Specifically, following [65], the responses were rescaled, centering by group mean using the following formula, where *i* indicates the subject and *j* the country: $X'_{ij} = \left(X_{i,j} - \overline{X}_j\right)$.

## 4. Results

Once the data were standardized to avoid the influence of cultural response bias, the Once the data were standardized to avoid the influence of cultural response bias, the adequate distribution of individuals across the different groups was confirmed (taking into account the gender and age variables), and the internal consistency of the measurement scales were verified, the ANOVA analysis could be performed using the statistic software STATISTICA 10.0. Prior to the analysis, the ANOVA assumptions were checked [66]. Specifically, Levene's test exceeded the value of 0.05 for all the tests performed, thus

confirming the homoscedasticity of variances. In addition, Mauchly's sphericity test (to validate that the variances of the differences between all possible group pairs are equal) was performed. This presented a *p*-value of 0.14, indicating that the assumption of sphericity was fulfilled [67]. Moreover, despite the fact that the distribution does not follow a normal distribution (Kolmogorov–Smirnov test; *p*-value 0.04), the sample size is more than sufficient to ensure the validity of the results obtained (for an experiment of this type, 2 × 2, the sample required for this would be approximately 120 individuals) [68]. Additionally, the statistical power was compared using the G-Power software 3.1, which indicated that, with a sample of 124 individuals, a statistical power of 0.9 would be achieved, taking into account an effect size of f = 0.25 [69].

The technology type (email vs. chatbot) and country (Czech Republic vs. Spain) were factors, and each of the dimensions of EQS and its overall evaluation was a dependent variable. Other studies in the tourism sphere have also used this type of technique to identify differences between groups (e.g., [70–72]).

The results of the ANOVA showed (Figure 2) an interaction effect between technology type and country for participants' overall evaluation of the quality of service received (ANOVA *p*-value < 0.01), without there being any main effect for either technology type (ANOVA *p*-value = 0.43) or country (ANOVA *p*-value = 0.65). In addition, arrows have been added to highlight the differences between each of the groups. Regarding the interaction effect, the participants from the Czech Republic (which has a relatively low-DL population) presented no significant differences in their evaluation of the two technologies (*p*-value for Tukey = 0.48). However, the Spanish participants (with higher DL) did present significant differences in favor of the chatbot (*p*-value for Tukey = 0.04). This suggests that the use of chatbots in tourism service provision will be valued positively by those source markets with higher DL, while tourists from countries with lower DL will be less likely to appreciate the use of this type of technology.

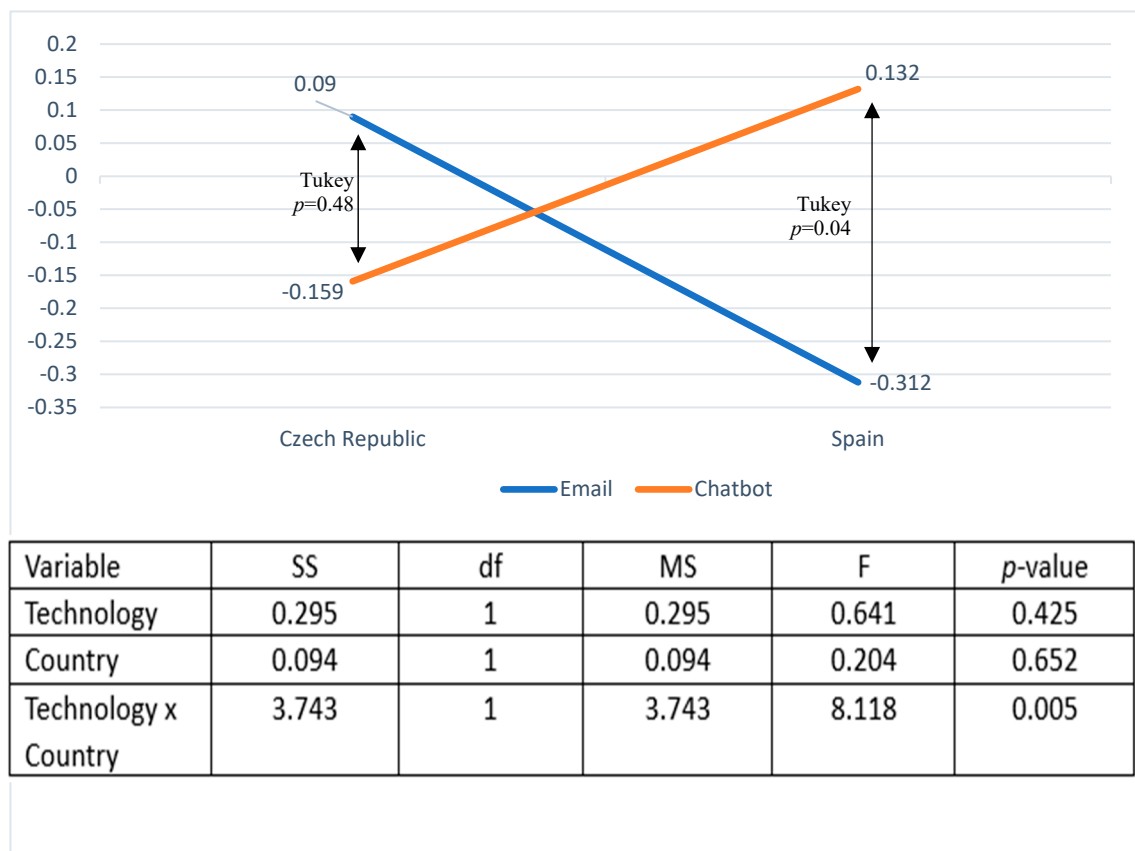

| Variable | SS | df | MS | F | *p*-value |
|---|---|---|---|---|---|
| Technology | 0.295 | 1 | 0.295 | 0.641 | 0.425 |
| Country | 0.094 | 1 | 0.094 | 0.204 | 0.652 |
| Technology x Country | 3.743 | 1 | 3.743 | 8.118 | 0.005 |

**Figure 2.** Interaction effect of technology type and country on user evaluation of ESQ.

Participants with a lower level of DL preferred email because they perceived it to provide greater security (Figure 3a; *p*-value = 0.01), empathy (Figure 3b; *p*-value = 0.00), reliability (Figure 3c; *p*-value = 0.03), and information quality (Figure 3f; *p*-value = 0.00). However, participants with higher DL preferred the chatbot due to its greater competence in completing the task (Figure 3d; *p*-value = 0.01). Finally, all the participants rated the responsiveness of the chatbot higher than that of email (Figure 3e; *p*-value = 0.01), bearing in mind that the email response was received no later than an hour after the request was sent by the participant. With that, we can confirm the proposed hypothesis H1A to H1F.

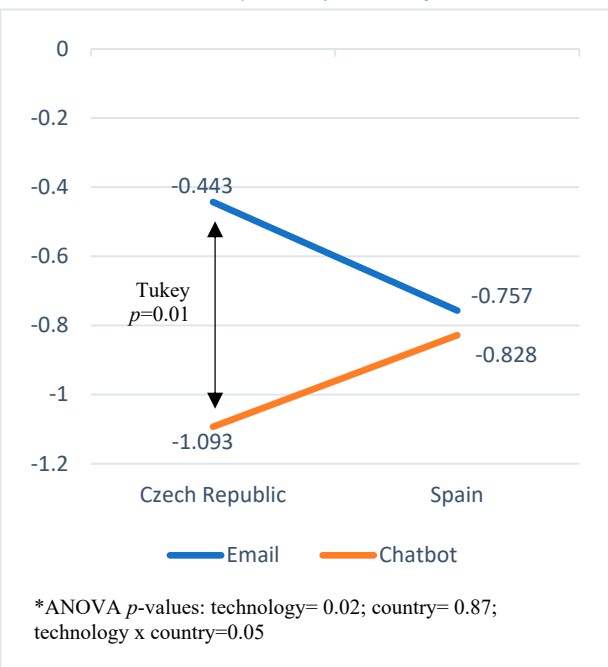

(**a**)

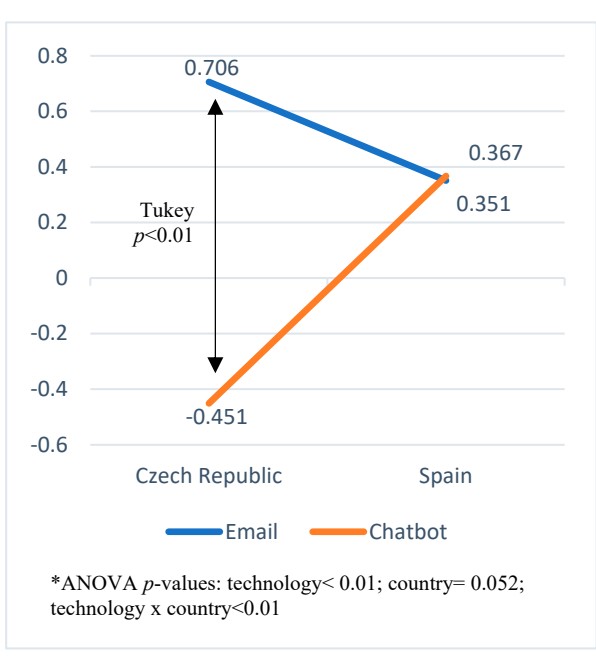

(**b**)

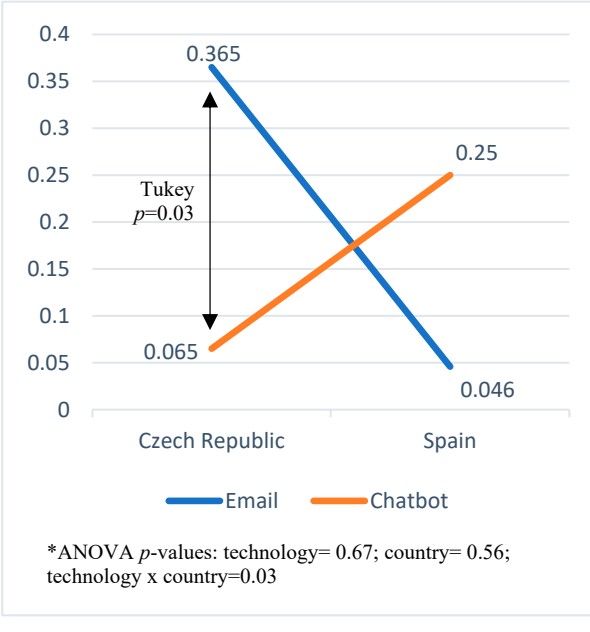

(**c**)

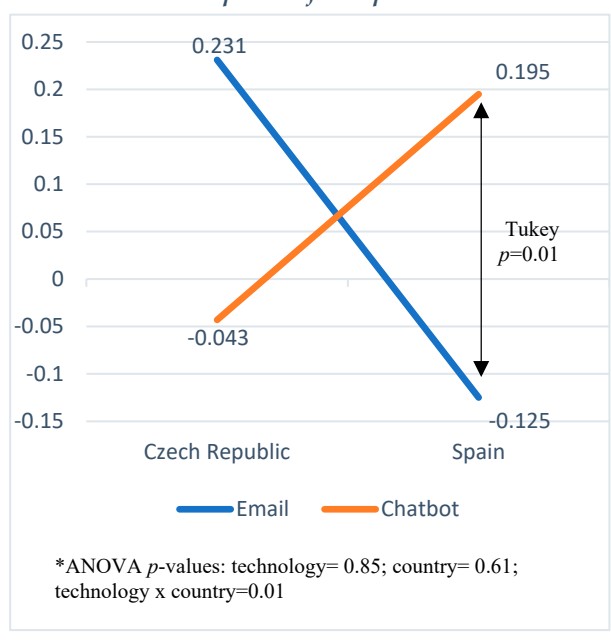

(**d**)

**Figure 3.** *Cont.*

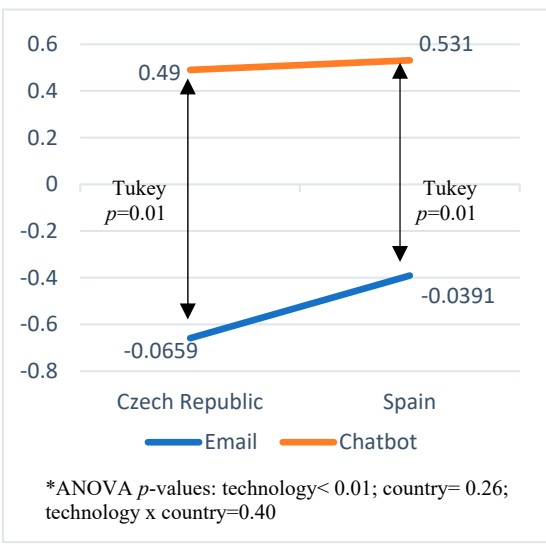

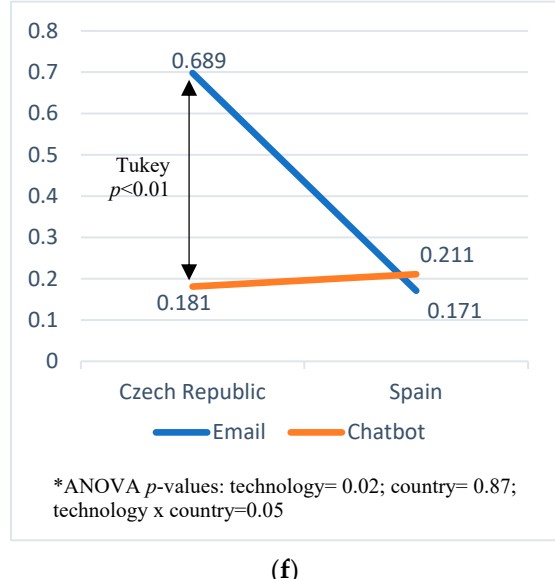

**(e)** **(f)**

**Figure 3.** Own elaboration. * ANOVA *p*-values for main and interaction effects are shown. Significant post hoc tests are indicated in the graphs.

An analysis of the results shows that DL affects individuals' evaluation of the quality of a technological service applied to tourism. This finding is in line with the literature, which suggests that DL affects new-technology use intention [18]. It also supports our research hypothesis that the individual's DL moderates their assessment of the quality of service delivered via the technology in question, such that a higher level of DL corresponds to a more favorable evaluation of the service. If we look at each dimension of e-service quality in countries with low DL, the present analysis identified significant differences in relation to security, empathy, reliability and information quality, depending on the technology used. This suggests that developers must endeavor to make significant progress in these aspects if this new technology (chatbot) is ever to replace existing ones and achieve a sound foothold in the tourism sector. However, one key factor in promoting technology use (both in countries with high DL and those with low DL) is responsiveness, which both groups of tourists rated superior in the case of the chatbot.

**5. Discussion**

Considering the different dimensions of electronic service quality, we can observe that, for the SC variable, countries characterized by lower DL regard a mature technology as more secure than a relatively new one [62], which may result in a poor perception of security in the case of a new technology. Thus, based on the hypothesis proposed in this work, we can confirm that DL does, indeed, act as a moderating element when it comes to perceiving the benefits of a new technology over a well-established one [16]. Furthermore, these differences in DL not only affect the user's ability to perceive the quality of a new technology, as per the findings of [13,58,62]. We also identified that, when the user presents a lower level of DL, they may not be able to perceive all the benefits that a new technology can offer, and therefore may not consider it to be reliable or useful in terms of information quality, thereby reducing their intention to use it and negatively affecting their perception of it [12,60]. Hence, developers and firms alike must factor in the heterogeneity of the market in which they are operating before selecting one technology over another [12].

Continuing with the dimensions of ESQ, it can be inferred from the present results that the empathy of a new technology is perceived to be lower in countries with low DL, compared to the greater perception of empathy in the case of a mature technology, which, according to the recent literature, will affect the user's ultimate satisfaction with

the technology [22]. Hence, including a human element in these conversational agents can increase their acceptance [28].

Turning to the last two dimensions, competence and responsiveness, the present results show that, in the case of the latter, there were differences according to technology type. That is, users perceived the chatbot to offer greater responsiveness than email, regardless of their level of DL. While the literature finds that DL affects perceptions of the benefits that a technology can offer [59] his can be compensated for if the respondents are digital natives [73], given that the chatbot's immediacy of response is a pivotal attribute for this cohort. Finally, with regard to competence, the results indicate that countries with higher DL regard the information provided by a mature technology as being of lower quality, while no differences were observed in the case of a country with lower DL. According to the literature, having a low level of DL may lead the user not to regard a new technology as useful [61].

In short, at present, completely substituting mature technologies (such as email) with chatbots for delivering online services to tourists remains unadvisable.

## 6. Conclusions and Limitations

Several conclusions can be drawn from these results that can contribute to the successful implementation of new technologies in the tourism sector. Beginning with the main management implications of this study, prior to any technological substitution, it is important to conduct a preliminary analysis of whether the target audience possesses the necessary skills, not only to use the technology correctly but also in terms of their ability to perceive it as superior to current or traditional alternatives [19]. In this sense, progressively introducing new technologies, to make them compatible over a more prolonged period, can help give tourists time to acquire the necessary skills. This progressive implementation can also encourage smaller firms in the sector to introduce these changes to their services little by little [12], which would render the sector in general more competitive and modern.

A further conclusion that can be drawn is that, when solving minor issues such as making a change to a hotel booking, a chatbot is regarded as being more responsive than email, regardless of the DL of the customer. Knowing this, hotel managers might consider transferring certain, less important services over to automated systems, such as the chatbot. Such a shift would bring with it the corresponding time savings for staff and cost savings for the firm, drawing on the advantages of a fast, no-wait service for users who wish to perform minor operations. However, more work still needs to be carried out to improve automated systems, given that, even when users present a relatively high level of DL, the information that such systems provide is not perceived to be superior to that delivered via a mature technology, nor are they perceived as more empathic systems—despite giving instant and personalized responses—or as more secure. Arguably, this latter aspect is the one that requires the greatest attention on the part of hotel firms if they seek to automate procedures that require sensitive customer data.

In addition to these conclusions on the managerial level, the findings of this study also hold theoretical implications. While previous works have already addressed the issue of ESQ [13,22], this study constitutes an advancement in that it compares two technologies: one that is fully embedded among the majority of clients (email) and another that is still in the process of being implemented by some firms in the sector (chatbot). This, together with the division of the sample into two publics that differ in their levels of DL, rendered it possible to demonstrate how DL acts as a moderator of the perceived quality of a new technology vs. one that is well established. Furthermore, these differences are demonstrated for each of the six dimensions of quality taken into account in this study. Regarding the theoretical contributions to the specific field of tourism, this research is, to the best of our knowledge, the first to capture these differences in perceptions of a chatbot system vs. another technology, which is of particular interest, given the recent boom in studies on the implementation of AI in the tourism sector. Other works in the sector have used this type

of technology in their studies, but not to compare the acceptance of two technologies with different levels of acceptance, such as email and chatbot (e.g., [55]).

Lastly, this work has certain limitations that need to be acknowledged, starting with the size of the sample, which included just two countries and was distinctly young in terms of age profile. Larger and more diverse samples would enable more generalizable results to be achieved. A further limitation is that we centered on above-average digital skills rather than intermediate ones. This decision was taken on the basis that, since the interviewees had to interact independently with the different technologies, it was important to choose the segment of the population that had the highest level of DL (university students). The other main limitation of this study is the design of the chatbot, which only had the capacity to perform one pre-programmed task. That said, the design was not so far-removed from that of current conversational agents in the sector, which still have a long way to go in terms of offering flexible conversations with consumers (that take into account human flaws in the use of language or the use of abbreviations or dialect) and learning to adapt their responses to user expectations. Finally, the number of individuals taken into account in our study is another limitation. Nevertheless, a test power of more than 0.9 was achieved with this sample number, which indicates that it is sufficient to have confidence in the results achieved.

**Supplementary Materials:** The following supporting information can be downloaded at: https://www.mdpi.com/article/10.3390/tourhosp5020018/s1, Supplementary File S1: Items for each dimension.

**Author Contributions:** Conceptualization, J.V.-O., J.A.C.-G. and J.B.; methodology, J.V.-O., J.A.C.-G. and J.B.; software, J.V.-O., J.A.C.-G. and J.B.; validation, J.V.-O., J.A.C.-G. and J.B.; formal analysis, J.V.-O., J.A.C.-G. and J.B.; investigation, J.V.-O., J.A.C.-G. and J.B.; resources, J.V.-O., J.A.C.-G. and J.B.; data curation, J.V.-O., J.A.C.-G. and J.B.; writing—original draft preparation, J.V.-O., J.A.C.-G. and J.B.; writing—review and editing, J.V.-O., J.A.C.-G. and J.B.; visualization, J.V.-O., J.A.C.-G. and J.B.; supervision, J.V.-O., J.A.C.-G. and J.B.; project administration, J.V.-O., J.A.C.-G. and J.B.; funding acquisition, J.V.-O., J.A.C.-G. and J.B. All authors have read and agreed to the published version of the manuscript.

**Funding:** This research was funded by MCIN/AEI/10.13039/501100011033 grant number PID2019-110941RB-I00.

**Institutional Review Board Statement:** Not applicable.

**Informed Consent Statement:** Not applicable.

**Data Availability Statement:** The datasets generated during and/or analyzed during the current study are available from the corresponding author upon reasonable request.

**Conflicts of Interest:** The authors declared no potential conflicts of interest with respect to the research, authorship, and/or publication of this article.

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
