# Peer review of "Chatbot Service Quality: An Experiment Comparing Two Countries with Different Levels of Digital Literacy"

_tourismhosp, doi:10.3390/tourhosp5020018_

Round 1

Reviewer 1 Report

Comments and Suggestions for Authors

Title: Chatbot Service Quality: An experiment comparing two countries with different levels of digital literacy

Comments

This study discusses an interesting new topic, tourists' digital literacy. The use of innovative and revolutionary new technologies in the tourism industry not only improves the quality of services, but also changes the way tourists consume, especially in terms of the acceptance and use of digital technologies by tourists, so the topic of tourists' digital literacy may be important and necessary. I only provide a few comments for the editors and authors' reference. 

#1Introduction

The authors could present more information about tourists' digital literacy, such as the importance of tourists' digital literacy in areas such as tourism service experience and evaluation and emphasize the need and urgency of research on tourists' digital literacy.

#2Literature review

The authors presented a definition of digital literacy but did not provide more information on the definition and structural dimensions of tourists' digital literacy. Therefore, I suggest that the authors conceptualize tourists' digital literacy based on the literature review.

#3Conclusion and limitation

The authors need to clarify and add to the theoretical contributions of this study.

Author Response

Comments:

Response:

This study discusses an interesting new topic, tourists' digital literacy. The use of innovative and revolutionary new technologies in the tourism industry not only improves the quality of services, but also changes the way tourists consume, especially in terms of the acceptance and use of digital technologies by tourists, so the topic of tourists' digital literacy may be important and necessary. I only provide a few comments for the editors and authors' reference. 

First of all, thanks for your considerations with our paper.

We the authors have tried to response all the comments.

#1:Introduction

The authors could present more information about tourists' digital literacy, such as the importance of tourists' digital literacy in areas such as tourism service experience and evaluation and emphasize the need and urgency of research on tourists' digital literacy.

Agree. We have clarified the gap and the aim of the study in the introduction section. Changes in page 2:

If we take into account that this technology is increasingly present in the tourism sector (Tussyadiah, 2020) and that more and more companies are opting to use them in different services offered to their customers, such as making or changing bookings, which implies a reduction in costs for the company and time for the tourist (Fan et al., 2022), it is necessary to know the users' opinion on the matter. To this end, some authors have been using some models of technology acceptance, such as the UTATU model to know if they are willing to use it (Gopinath and Kasilingam, 2023). These works on technology uptake use digital literacy to look at the level of technology acceptance, however a different level of digital literacy may also imply different perceptions of perceived quality, which may further hinder uptake in certain regions (Castañeda et al., 2020; Cheng et al., 2022), however, this is the first work, to the best of our knowledge, that attempts to determine the factors that affect the introduction of chatbots into the tourism sector, studying the quality of service offered by an already established technology such as email versus a new one such as the chat bot. The results can help tourism managers to carry out a correct implementation of these technologies, since it takes into account the differences in the perception of the quality of service offered by each one for subjects with different levels of digital literacy.

New references:

Castañeda, J. A., Frías-Jamilena, D. M., Rodríguez-Molina, M. A., & Jones, A. (2020). Online Marketing Effectiveness-the influence of information load and digital literacy, a cross-country comparison. Electronic Markets, 30, 759-773.

Cheng, C., Gearon, E., Hawkins, M., McPhee, C., Hanna, L., Batterham, R., & Osborne, R. H. (2022). Digital health literacy as a predictor of awareness, engagement, and use of a national web-based personal health record: population-based survey study. Journal of Medical Internet Research, 24(9), e35772.

Fan, X., Jiang, X., and Deng, N. (2022). Immersive technology: A meta-analysis of augmented/virtual reality applications and their impact on tourism experience. Tourism Management, 91, 104534.

Gopinath, K., and Kasilingam, D. (2023). Antecedents of intention to use chatbots in service encounters: A meta‐analytic review. International Journal of Consumer Studies.

Tussyadiah, I. (2020). A review of research into automation in tourism: Launching the Annals of Tourism Research Curated Collection on Artificial Intelligence and Robotics in Tourism. Annals of Tourism Research, 81, 102883.

#2:Literature review

The authors presented a definition of digital literacy but did not provide more information on the definition and structural dimensions of tourists' digital literacy. Therefore, I suggest that the authors conceptualize tourists' digital literacy based on the literature review.

Agree. Thanks for your comment. We have tried to highlighted this concept in tourism related literature. Changes in the page:

In the tourism sector, some studies have used this construct to explain the perceived ease of use of some techonoligies in their travels (Liu et al., 2017), by including in the acceptance technology models (Bae and Han, 2020), linking the DL with the motivation to obtain autonomy during travel (Tussyadiah and Wang, 2016) or to eplain the perception of the congruency between digital app presentations and their actual experiences (Xiong and Zhang, 2024). DL has been used over the years within these technology acceptance models, such as TAM or UTAUT in the tourism sector (Xiong and Zhang, 2024).

New references;

Bae, S. Y., & Han, J. H. (2020). Considering cultural consonance in trustworthiness of online hotel reviews among generation Y for sustainable tourism: An extended TAM model. Sustainability, 12(7), 2942.

Liu, Y., Li, K. J., Chen, H., & Balachander, S. (2017). The effects of products’ aesthetic design on demand and marketing-mix effectiveness: The role of segment prototypicality and brand consistency. Journal of Marketing81(1), 83-102.

Tussyadiah, I. P., & Wang, D. (2016). Tourists’ attitudes toward proactive smartphone systems. Journal of Travel Research55(4), 493-508.

Xiong, S., & Zhang, T. (2024). Enhancing tourist loyalty through location-based service apps: Exploring the roles of digital literacy, perceived ease of use, perceived autonomy, virtual-content congruency, and tourist engagement. Plos one19(1), e0294244.

#3:Conclusion and limitation

The authors need to clarify and add to the theoretical contributions of this study.

Again, thanks for your comments.

We have clarified this issue in the conclusion sector. Changes in the page 11:

In addition to these conclusions on the managerial level, the findings of the study also hold theoretical implications. While previous works have already addressed the issue of ESQ (Hanna, 2018; Zouari and Abdelhedi, 2021), this study constitutes an advancement in that it compares two technologies: one that is fully embedded among the majority of clients (email) and another that is still in the process of being implemented by some firms in the sector (chatbot). This, together with the division of the sample into two publics that differ in their levels of DL, rendered it possible to demonstrate how DL acts as a moderator of the perceived quality of a new technology vs. one that is well established. Furthermore, these differences are demonstrated for each of the six dimensions of quality taken into account in the study. Regarding the theoretical contributions to the specific field of tourism, this research is, to the best of our knowledge, the first to capture these differences in perceptions of a chatbot system vs. another technology, which is of particular interest, given the recent boom in studies on the implementation of AI in the tourism sector (e.g., Bigné and Matamura, 2023). Other works in the sector have used this type of technology in their studies, but not to compare the acceptance of two technologies with different levels of acceptance, such as email and chatbot. (e.g. Bae and Han, 2020; Xiang and Zhang., 2024).

Reviewer 2 Report

Comments and Suggestions for Authors

The paper is very interesting. It is publishable in present form.

Author Response

Thank you very much for your consideration of our work. 

Reviewer 3 Report

Comments and Suggestions for Authors

Dear respected authors,

The content of the manuscript must be improved. Please consider the following remarks:

  1. The content of the abstract section is quite fine. However, there is a sentence, and I am curious whether it is literally and scientifically correct. In the sentence “Finally, all the participants rated the responsiveness of the chatbot higher than that of email,” The word “all” should be checked, because if all the respondents had such an idea, the data gathering—I mean participant selection—and consequently the data used were biased. Otherwise, please correct the sentence.
  2. Keywords should be selected based on their importance and frequent repetition in the text. Hence, the keywords "international tourism” and “cross-cultural analysis” should be eliminated or modified as they have not been used in the text.
  3. Accessibility to the data set and the declaration of authors should be moved to the conclusion section.
  4. The research gap and the aim of the study should be highlighted, and their coherence should be supported.
  5. It is expected to see the hypothesis considered in this study based on the provided research model.
  6. In Section 3.2, it has been mentioned that “mean age (p-value = 0.54) and the percentage of women (p-value = 0.58)"  should be less than or equal to 0.05 or 0.01 to decide whether to reject or not to reject a hypothesis. Please explain the reason for using this parameter or modify it.
  7. There are some sentences like “Graph Error! No text of specified style in the document” in Section 4 should be corrected.
  8. It seems the respected authors used Excel software to test the hypothesis and for the descriptive statistics. It is highly recommended to use scientific statistical software to analyze the data.
  9. Figure numbers and titles are not provided.
  10. There are some arrows on the graphs that were not explained.

Regards

Author Response

Comments

Response

The content of the manuscript must be improved. Please consider the following remarks:

First of all, thanks for your valuable comments which with no doubt have improved our paper.

The content of the abstract section is quite fine. However, there is a sentence, and I am curious whether it is literally and scientifically correct. In the sentence “Finally, all the participants rated the responsiveness of the chatbot higher than that of email,” The word “all” should be checked, because if all the respondents had such an idea, the data gathering—I mean participant selection—and consequently the data used were biased. Otherwise, please correct the sentence.

Agree. We have changed this sentence by eliminating the word all, thanks for your appreciation.

We want to refer that it is found that “participants rated the responsiveness of the chatbot higher than that of email”.

Keywords should be selected based on their importance and frequent repetition in the text. Hence, the keywords "international tourism” and “cross-cultural analysis” should be eliminated or modified as they have not been used in the text.

Agree. Thanks for your valuable comment.

We have erased these two keywords and replace for:

Keywords: Technology acceptance; Digital literacy; e-Service Quality; Automated services; Technology in tourism

Accessibility to the data set and the declaration of authors should be moved to the conclusion section.

Agree.

We have moved it at the end of the mentioned section.

The research gap and the aim of the study should be highlighted, and their coherence should be supported.

Agree. We have clarified the gap and the aim of the study in the introduction section. Changes in page 2:

If we take into account that this technology is increasingly present in the tourism sector (Tussyadiah, 2020) and that more and more companies are opting to use them in different services offered to their customers, such as making or changing bookings, which implies a reduction in costs for the company and time for the tourist (Fan et al., 2022), it is necessary to know the users' opinion on the matter. To this end, some authors have been using some models of technology acceptance, such as the UTATU model to know if they are willing to use it (Gopinath and Kasilingam, 2023). These works on technology uptake use digital literacy to look at the level of technology acceptance, however a different level of digital literacy may also imply different perceptions of perceived quality, which may further hinder uptake in certain regions (Castañeda et al., 2020; Cheng et al., 2022), however, this is the first work, to the best of our knowledge, that attempts to determine the factors that affect the introduction of chatbots into the tourism sector, studying the quality of service offered by an already established technology such as email versus a new one such as the chat bot. The results can help tourism managers to carry out a correct implementation of these technologies, since it takes into account the differences in the perception of the quality of service offered by each one for subjects with different levels of digital literacy.

New references:

Castañeda, J. A., Frías-Jamilena, D. M., Rodríguez-Molina, M. A., & Jones, A. (2020). Online Marketing Effectiveness-the influence of information load and digital literacy, a cross-country comparison. Electronic Markets, 30, 759-773.

Cheng, C., Gearon, E., Hawkins, M., McPhee, C., Hanna, L., Batterham, R., & Osborne, R. H. (2022). Digital health literacy as a predictor of awareness, engagement, and use of a national web-based personal health record: population-based survey study. Journal of Medical Internet Research, 24(9), e35772.

Fan, X., Jiang, X., and Deng, N. (2022). Immersive technology: A meta-analysis of augmented/virtual reality applications and their impact on tourism experience. Tourism Management, 91, 104534.

Gopinath, K., and Kasilingam, D. (2023). Antecedents of intention to use chatbots in service encounters: A meta‐analytic review. International Journal of Consumer Studies.

Tussyadiah, I. (2020). A review of research into automation in tourism: Launching the Annals of Tourism Research Curated Collection on Artificial Intelligence and Robotics in Tourism. Annals of Tourism Research, 81, 102883.

It is expected to see the hypothesis considered in this study based on the provided research model.

Agree.

We have included sub-hypothesis for each dimension. Changes in the results section have been made consequently. Changes in the manuscript. Page 5:

H1: DL moderates the user’s perception of the perceived quality of a new technology compared to a mature one.

H1A: the perception of security will differ from a new technology compared to a mature one.

H1B: the perception of empathy will differ from a new technology compared to a mature one.

H1C: the perception of reliability will differ from a new technology compared to a mature one.

H1D: the perception of competence will differ from a new technology compared to a mature one.

H1E: the perception of responsiveness will differ from a new technology compared to a mature one.

H1F: the perception of information quality will differ from a new technology compared to a mature one.

In Section 3.2, it has been mentioned that “mean age (p-value = 0.54) and the percentage of women (p-value = 0.58)"  should be less than or equal to 0.05 or 0.01 to decide whether to reject or not to reject a hypothesis. Please explain the reason for using this parameter or modify it.

The test that we have carried out is a test of mean differences to check that both groups are stable. In this test the null hypothesis states that there is equality of means. It has been specified in the text.

Changes in the manuscript. Page 6:

These data were balanced across the two technologies under analysis, with no differences in either the mean age (p-value=0.54) or in the percentage of women (p-value=0.58) (where H0 means that there are no differences in the average by groups). By country, no differences were identified in the socio-demographic variables, with p-values of 0.69 and 0.11, respectively, for age and gender.

There are some sentences like “Graph Error! No text of specified style in the document” in Section 4 should be corrected.

Agree.

We have made changes. It was probably a problem with the format.

It seems the respected authors used Excel software to test the hypothesis and for the descriptive statistics. It is highly recommended to use scientific statistical software to analyze the data.

We used Excel for the graphs. To conduct the ANOVA analysis, we used STATISTICA. We have clarified this in the paper. Thanks for your comment. Changes in the manuscript, page 6:

Once the data had been standardized to avoid the influence of cultural response bias, the adequate distribution of individuals across the different groups had been confirmed (taking into account the gender and age variables), and the internal consistency of the measurement scales had been verified, the ANOVA analysis could be performed using the statistic software STATISTICA.

Figure numbers and titles are not provided.

Regarding this issue we can see the figure numbers and titles. We do not know if there is a problem with the sending document.

Here you can find the summary:

Figure 1. Dimensions of ESQ.

Table 1. Sample data.

Graph 1. Interaction effect of technology type and country on user evaluation of ESQ.

Graph 2.1 Perception of security

Graph 2.2 Perception of empathy

Graph 2.3. Perception of reliability

Graph 2.4. Perception of competence

Graph 2.5 Perception of responsiveness

Graph 2.6. Perception of information quality

Additionally, some mistakes have been found in the figure 1, that the authors have fixed.

There are some arrows on the graphs that were not explained

Agree.

We have included the following explanation in the text in order to clarify the meaning of these arrows. Page number 7:

In addition, arrows have been added to highlight the differences between each of the groups.

Reviewer 4 Report

Comments and Suggestions for Authors

1.      What is the main question addressed by the research?

The study compares a more innovative and interactive service-provision technology (a chatbot) with a more traditional one (email). To this end, an experiment was conducted in which 124 participants from Spain (higher DL) and the Czech Republic (lower DL).

The intention is original and interesting. Although the sample is unfortunately small and it is difficult to understand the justification for choosing the countries it comes from.

2.      Do you consider the topic original or relevant in the field? Does it
address a specific gap in the field?

The study corresponds to the journal's profile. The research is aimed at acquiring new knowledge and using it in the practice of hotel guest service. In the practical part, the research is certainly original and creative.

3.      What does it add to the subject area compared with other published
material?

Research results indicate that the assessment of the quality of new hotel technologies depends on the user's digital competences. This can be treated as a new, practically valuable research insight.

4.      What specific improvements should the authors consider regarding the
methodology? What further controls should be considered?

The methodology is correctly selected and used. I have no comments.

5.      Are the conclusions consistent with the evidence and arguments presented
and do they address the main question posed?

The article is written in a clear way. The goal corresponds to research and research results.

6.      Are the references appropriate?

The literature review is based on well-selected literature. A very good approach is to divide it into individual aspects of using new technologies in tourism. The literature does not require supplementation.

7.       Please include any additional comments on the tables and figures.

The graphic material is sufficient, well selected and correctly described.

Before publishing, it should be clearly defined what new research results bring to science in terms of the acceptance of new technologies. How can they be used to improve services in the hotel industry? These threads seem to be insufficiently developed yet.

Comments on the Quality of English Language

English is OK.

Author Response

Reviewer 4:

First of all, we would like to thank you for your valuable work and time spent on our work.

We, the authors, have tried to respond as well as possible to your indications, hopefully to your pleasure:

  1. What is the main question addressed by the research?

The study compares a more innovative and interactive service-provision technology (a chatbot) with a more traditional one (email). To this end, an experiment was conducted in which 124 participants from Spain (higher DL) and the Czech Republic (lower DL).

The intention is original and interesting. Although the sample is unfortunately small and it is difficult to understand the justification for choosing the countries it comes from.

 Agree. It was included as a limitation:

Changes in the manuscript:

Finally, the number of individuals taken into account in our study is another limitation. Nevertheless, a test power of more than 0.9 was achieved with this sample number, which indicates that it is sufficient to have confidence in the results achieved.

Furthermore, some analysis of the reliability of our results have been made in order to demonstrate the robustness of our results:

“Prior to the analysis, the ANOVA assumptions were checked (Christensen, 2018). Specifically, Levene’s test exceeded the value of 0.05 for all the tests performed, thus confirming the homoscedasticity of variances. In addition, Mauchly’s sphericity test (to validate that the variances of the differences between all possible group pairs are equal) was performed. This presented a p-value of 0.14, indicating that the assumption of sphericity was fulfilled (Miller, 1997). Moreover, despite the fact that the distribution does not follow a normal distribution (Kolmogorov-Smirnov test; p-value 0.04), the sample size is more than sufficient to ensure the validity of the results obtained (for an experiment of this type, 2x2, the sample required for this would be approximately 120 individuals) (Tabachnick and Fidell, 2007). Additionally, the statistical power was compared using G-Power software, which indicated that, with a sample of 124 individuals, a statistical power of 0.9 would be achieved, taking into account an effect size of f=0.25 (Karadag and Aktas, 2012).”

Before publishing, it should be clearly defined what new research results bring to science in terms of the acceptance of new technologies. How can they be used to improve services in the hotel industry? These threads seem to be insufficiently developed yet.

Agree. We have tried to improve in the introduction section the scientific and managerial originality that our paper can offer. Changes in the manuscript:

If we take into account that this technology is increasingly present in the tourism sector (Tussyadiah, 2020) and that more and more companies are opting to use it in different services offered to their customers, such as making or modifying reservations, which means a reduction in costs for the company and time for the tourist (Fan et al., 2022), it is necessary to know the users' opinion on the matter. For this purpose, some authors have been using some technology acceptance models, such as the UTAUT model, to find out whether they are willing to use it (Gopinath and Kasilingam, 2023). These works on technology acceptance use digital literacy to analyse the level of technology acceptance, however, a different level of digital literacy may also imply different perceptions of perceived quality, which may further hinder acceptance in certain regions (Castañeda et al., 2020; Cheng et al., 2022), however, this is the first work, to our knowledge, that attempts to determine the factors affecting the introduction of chatbots in the tourism sector by studying the quality of service offered by an established technology such as email versus a new one such as chat bot. This can mean cost savings for companies but also time savings for tourists, thus increasing their satisfaction with the destination (Vena-Oya et al., 2021). Thus, firstly, the scientific novelty of this work lies in the use of the perception of quality of service in countries with different levels of digital literacy (Spain vs. Czech Republic). This represents an advance in terms of understanding how these systems should be correctly implemented in a sector as important for the economy of countries as tourism, given that, as not all countries perceive it as better (the Czechs see some aspects of email as better), this implementation should not be generalised and managers should be staggered according to the digital readiness of citizens and tourists in each country.

New references:

Vena-Oya, J., Castaneda-García, J. A., Rodríguez-Molina, M. A., & Frías-Jamilena, D. M. (2021). How do monetary and time spend explain cultural tourist satisfaction?. Tourism Management Perspectives, 37, 100788.

Round 2

Reviewer 1 Report

Comments and Suggestions for Authors

This paper makes significant progress in clarifying the entire study and becoming more precise.

Authors and proofreaders should read the entire manuscript, including tables and figures (not just the revised portion), to ensure accuracy and consistency.

Comments on the Quality of English Language

The article still has some grammatical errors that need to be corrected; such obvious errors seriously damage the quality of the text.

Author Response

Thank you very much for your time and valuable comments. It has been an honour for us to have your experience in this review process.

The paper has been proofread by a native English speaker who has tried to correct any possible errors.

Reviewer 3 Report

Comments and Suggestions for Authors

Dearly respected authors,

The manuscript's content has been modified according to what the reviewer recommended. According to the reviewer's point of view, the manuscript is worth publishing in the respected journal.

Regards,

Author Response

Thank you very much for your time and valuable comments. It has been an honour for us to have your experience in this review process. 

Reviewer 4 Report

Comments and Suggestions for Authors

The article was corrected according my suggestions. It could be published now.